# The Effects of a Single Session of a Rhythmic Movement Program on Selected Biopsychological Parameters in PD Patients: A Methodological Approach

**DOI:** 10.3390/medicina59081408

**Published:** 2023-08-01

**Authors:** Claire Chrysanthi Karpodini, Themistoklis Tsatalas, Ioannis Giannakopoulos, Mattias Romare, Giannis Giakas, Panagiotis V. Tsaklis, Petros C. Dinas, Aline Nogueira Haas, Sokratis G. Papageorgiou, Efthalia Angelopoulou, Matthew A. Wyon, Yiannis Koutedakis

**Affiliations:** 1Sport and Physical Activity Research Centre, Faculty of Education, Health and Wellbeing, University of Wolverhampton, Wolverhampton WV1 1LY, UK; m.wyon@wlv.ac.uk; 2Department of Physical Education and Sport Science, University of Thessaly, 421 00 Trikala, Greece; ttsatalas@gmail.com (T.T.); ptsaklis@gmail.com (P.V.T.);; 3Department of Molecular Medicine and Surgery, Karolinska Institute, 171 65 Solna, Sweden; 4School of Physical Education Physiotherapy and Dance, Federal University of Rio Grande do Sul (UFRGS), Porto Alegre 91410-000, Brazil; 5First Department of Neurology, Medical School, National and Kapodistrian University of Athens, Eginition University Hospital, 115 28 Athens, Greece

**Keywords:** rhythm, musicokinetic, dance, Parkinson’s disease, anxiety, gait disturbances, walking kinematics, walking kinetics, dorsolateral prefrontal cortex hemodynamic

## Abstract

The aim of the present study is to examine the acute effects of a specially designed musicokinetic (MSK) program for patients with Parkinson’s disease (PD) on (a) anxiety levels, (b) select kinematic and kinetic parameters, and (c) frontal cortex hemodynamic responses, during gait initiation and steady-state walking. *Methods*: This is a blind cross-over randomized control trial (RCT) in which 13 volunteers with PD will attend a 45 min MSK program under the following conditions: (a) a synchronous learning format and (b) an asynchronous remote video-based format. Changes in gait biomechanics and frontal cortex hemodynamic responses will be examined using a 10-camera 3D motion analysis (Vicon T-series, Oxford, UK), and a functional near-infrared spectroscopy (f-NIRS-Portalite, Artinis NL) system, respectively, while anxiety levels will be evaluated using the Hamilton Anxiety Rating Scale. *Expected results*: Guided by the rules of music, where periodicity is distinct, our specially designed MSK program may eventually be beneficial in improving motor difficulties and, hence, reducing anxiety. The combined implementation of f-NIRS in parallel with 3D gait analysis has yet to be evaluated in Parkinsonian patients following a MSK intervention. It is expected that the aforementioned intervention, through better rhythmicity, may improve the automatization of motor control, gait kinematics, and kinetics—supported by decreased frontal cortex hemodynamic activity—which may be linked to reduced anxiety levels.

## 1. Introduction

Parkinson’s disease (PD) is a neurodegenerative disorder with increasing rates in Western civilization. According to the World Health Organization, the prevalence of the disease has increased by 81% since 2000 [1].

Increased levels of anxiety, gait impairments, and lack of coordination are common non-motor and motor symptoms in PD, and can seriously impact patients’ overall quality of life. For instance, anxiety is a common psychiatric commonality in PD with symptoms occurring in between 20 and 52% of patients, while motor fluctuations—which are often accompanied by alterations in non-motor symptoms—can be seen in up to 75% of PD patients [2]. It has been found that fluctuations of non-motor parameters are considered risk factors for anxiety [3], and are associated with worse subjective motor symptoms, more severe gait problems, dyskinesia, and/or episodes of freezing [2,4]. However, although anxiety seems to be an important manifestation of PD, there is a lack of research regarding non pharmacological treatments [4].

As anxiety is associated with mobility difficulties in this population [2], it can be hypothesized that the lack of controlled rhythmic movement may be a factor exacerbating the condition. Gait impairment and lack of coordination are also associated with the disruption of patients’ internal rhythm, as dysfunction of basal ganglia has been linked to uncoordinated movement patterns [5,6]. Inner rhythm itself comprises the organizational characteristic of human movement reflected by its repetitive nature, and affects critically important elements of quality of life [7].

The potential mechanistic link between increased anxiety and gait-impairment severity of PD patients may be two-fold: Firstly, through increased conscious motor control. It has been previously hypothesized that decreases in automatization and rhythmicity, and their corresponding detrimental effects on gait, may be compensated for by enhanced conscious motor control. This process is mediated by increased activation of the pre-frontal cortex and, in particular, the dorsolateral pre-frontal cortex (DLPFC) [8]. Secondly, through detrimental effects of increased anxiety and rumination on executive functioning and working memory [9,10]. Based on this concept, if PD patients preferentially utilize executive motor control during the gait, the activation of these areas (i.e., DLPFC) would be enhanced, and higher anxiety levels would subsequently impair executive function, upon which gait is greatly dependent.

Humans have the tendency to synchronize their steps with the beat of the music they are listening to [11]. This phenomenon is associated with rhythm, the primal aspect of music, highlighting the close link between rhythm, movement, and dance as fundamental aspects of human existence [11]. Research in PD has revealed that rhythmic training seems to improve synchronization of neural activity in the brain, especially in the basal ganglia [12]. A systematic review evaluating the effect of rhythmic auditory stimulation on motor and non-motor parameters in PD patients revealed positive effects on gait parameters and mobility [13]. Another study revealed that rhythmic movement interventions that incorporate visual and auditory cues can improve sensorimotor integration [14]. However, rhythmics or musicokinetic education have not been examined in patients with PD. 

Rhythmics or musicokinetic education considers the human as a physical mental and spiritual entity. It is based on Francois Delsarte’s (1811–1871) and Jaque Dalcroze’s eurhythmics method (JDE), while more kinetic elements have been added throughout the years, especially for dance purposes, without neglecting the rules of music [15]. The term musicokinetic training (MSK for PD) is used herein. 

The aim of the proposed study is to examine the acute effects of a MSK program on select kinematic and kinetic parameters alongside DLPFC hemodynamics during gait initiation and steady-state walking in individuals with PD, performed in the following conditions: (a) a synchronous learning format and (b) an asynchronous remote video-based format. The program is customized for PD patients without deviating from the objectives of musicokinetic education. Primary outcomes will be the acute effects of a MSK protocol in people with PD on anxiety and locomotion. Secondary outcomes will be the effectiveness of an asynchronous remote video-based format for those patients that are not able to attend classes on site.

The novelty of the current design is the combined implementation of functional near-infrared spectroscopy (f-NIRS) in parallel with 3D gait analysis, which, until now, has yet to be evaluated in Parkinsonian patients following a MSK intervention. f-NIRS is extensively employed in chronic neurological disorders associated with aging, enabling direct measurements of the absolute concentrations of oxygenated and deoxygenated hemoglobin in the brain [16]. In relation to Parkinson’s disease, f-NIRS investigates the activity of the prefrontal cortex during tasks such as gait, postural stability, and dual-task performance, thereby determining motor vs. non-motor effects and executive dysfunction due to the disease’s pathology [16]. We shall non-invasively evaluate DLPFC hemodynamic activity in Brodmann area 46 during gait kinematics and kinetics evaluation using f-NIRS in order to investigate the cognitive involvement. Moreover, we argue that f-NIRS in conjunction with 3D gait analysis may provide valuable insights for understanding the mechanisms underlying the benefits of MSK program for PD.

## 2. Materials and Methods

### 2.1. Participants

Patients from central Greece diagnosed with PD will be asked to volunteer. PD diagnosis will be connected to certified neurologists, according to Movement Disorders Society (MDS) clinical diagnostic criteria. The campaign to attract volunteers will be organized by the Department of Physical Education and Sport Science, University of Thessaly, Greece, the local Medical Association, local authorities, and social media.

### 2.2. Inclusion and Exclusion Criteria

Participant inclusion criteria: (1) Hoehn and Yahr (H&Y) Parkinson disease scale I-III, (2) up to 75 years of age, (3) no restriction on disease duration, gender, or type of drug therapy, except for stable antiparkinsonian medication of at least 6 months, (4) data collections “on” medication, (5) no participation in any exercise program for at least 2 months, and (6) 100% attendance of the session. Exclusion criteria will include patients with dementia, cancer, cardiovascular diseases, poor visual or auditory capability and musculoskeletal problems, stable antiparkinsonian therapy, and patients with deep brain stimulation.

### 2.3. Sample Size Calculation

The total sample size was estimated through an a priori power analysis, using the G power V 3.1.9.7 software (Heinrich-Heine-Universität, Düsseldorf, Germany). The following input parameters were applied using a repeated-measure ANOVA design: effect size f = 0.33, α = 0.05, power = 0.80, and correlation between repeated measures r = 0.50. The a priori power calculation revealed that the initial sample size required is 11. By considering a 10% dropout rate, 13 individuals with Parkinson’s disease will be recruited to voluntarily participate in this study.

### 2.4. Experimental Design

A cross-over randomized control trial (RCT) will be adopted to investigate the acute effect of the MSK program for PD on anxiety, selected kinematic and kinetic parameters, and frontal cortex hemodynamic responses during gait initiation and during steady-state walking in people with PD, under the following conditions: (a) a synchronous learning format and (b) an asynchronous remote video-based format.

Volunteers will be asked to attend both conditions in a randomized order, separated by a washing-out period of at least one week. Levels of anxiety, kinematic, and DLPFC hemodynamic activity data will be collected before and after the two different interventions, which will last for approximately 45 min. The control situation will be defined based on the data collection at the same time points as above, but without any intervention following the 1-week washing out period. Ethical approval has been granted by Ethics Committee of University of Thessaly-Protocol number 2007. All procedures will be conducted in accordance with the declaration of Helsinki, in an appropriately equipped and safe laboratory, by qualified researchers with relevant experience. Research data are confidential and available to participants, who will be allowed to stop their participation at any time without previous notice.

### 2.5. Randomization and Blinding

Participants will be randomly (randomizer.org) assigned by a leading researcher blinded to the study either to start with the synchronous intervention or with the asynchronous remote video-based intervention. Two evaluators also blinded to the scope of the study will take all measurements before and after interventions. It will not be possible to blind participants as the delivery of the class will reveal their allocation.

### 2.6. Intervention

All intervention classes, synchronous and asynchronous remote video-based, will be held in the Biomechanics & Ergonomics Laboratory (b) at the Department of Physical Education and Sport Science, University of Thessaly, Trikala, Greece. Their structure is focusing on refining innate musicality and coordination via rhythmic movement, hearing training and improvisation.

Risk management strategy includes a qualified carer who will be present during both the synchronous and asynchronous modalities, full explanation of the experimental procedures, controlled laboratory temperatures at about 22° Celsius, as well as the continuous presence of at least two members of the research team.

The protocol has a duration of 45 min of rhythmic training and includes sections of warming up, stretching, breathing, movement combinations, a dance sequence, and recovery. Each exercise is accompanied by music that provides a clear and predictable rhythm. More specifically, the beat or rhythm of the music is used as a guide for the timing and duration of specific movements, allowing people with PD to coordinate their movements with the music. Simple values will be introduced, such as whole notes, half notes, and quarter notes, clapping and tapping, breathing, and sounds. Class will be accompanied by recorded classical piano music to meet the needs of the synchronous learning format and asynchronous remote video-based format. The intensity of the class will be moderate, at approximately 3 MET. The instructor will be an experienced dance teacher in musicokinetic education. Protocol details can be found in Table 1.

### 2.7. Outcome Measures

Volunteers will attend both conditions (synchronous and asynchronous remote video-based) in a randomized order, separated by a washing-out period of at least a week. Kinematic, DLPFC hemodynamic activity and levels of anxiety data will be collected before (base-line data) and after the two interventions, while, on a third occasion, control data will be collected at the same time points but without any intervention.

Two experienced evaluators blinded to the group allocation will collect data. Baseline data collection will include: (1) Anthropometric characteristics (sex, age, body mass index), H&Y scale; (2) Anxiety Hamilton Rating Scale (Hamilton 1960); (3) Gait analysis using a 10-camera 3D motion analysis system (Vicon T-series, Oxford, UK) and two force platforms (Bertec 4060–10, OH) embedded in the laboratory floor; (4) Frontal cortex hemodynamic responses using f-NIRS (Portalite by Artinis NL); and (5) Unified Parkinson’s Disease Assessment Scale-III (UPDRS-III). Post intervention measurements will include: (1) Anxiety Hamilton Rating Scale (Hamilton 1960); (2) Gait analysis using a 10-camera 3D motion analysis system (Vicon T-series, Oxford, UK) and two force platforms (Bertec 4060–10, OH) embedded in the laboratory floor; (3) Frontal cortex hemodynamic responses using f-NIRS (Portalite by Artinis NL); and (4) Unified Parkinson’s Disease Assessment Scale-III (UPDRS-III).

### 2.8. Primary Outcome Measures

Levels of anxiety will be evaluated via Hamilton Anxiety Rating Scale [17]. Ground reaction force (GRF) data captured at 1000 Hz for the right and left leg by the force platforms will be synchronized with the kinematic data obtained by the Vicon motion analysis system at 100 Hz. Twenty reflective markers will be placed bilaterally on the pelvis and lower extremities according to the marker set described in the literature [18,19]. The symmetrical center of rotation estimation (SCoRE) [20] and the symmetrical axes of rotation approach (SARA) [21] will be applied to estimate the hip joint center and the knee joint flexion axis, respectively.

For evaluation of DLPFC activation during gait, non-invasive measurements of local cortical hemoglobin (hBT) (i.e., blood volume), deoxyhemoglobin (HbH), and oxyhemoglobin (HbT) will be executed using a portable spectrometer (f-NIRS: PortaLite mini, Artinis Medical Systems, Zetten, Netherlands) with a capacity of light emission and detection of 840–760 nm, using a sampling frequency of 10 hz. The distance between NIRS optodes (i.e., sender and receiver) will be 25 mm, enabling a maximum penetration depth of 23 mm and an average measurement depth of 12.5 mm. Increased brain activity is generally considered to mirror increases in local HbO_2_ and decreases in HHb, resting on the mechanistic foundation of neurovascular coupling [22]. Decreased HbO_2_ responses can be considered a measure of neuronal suppression or altered hemodynamic re-distribution. The f-NIRS probe will be placed on the PFC of the right DLPFC [23]. To suppress any confounding influence of light, the probe shall be covered using black tape and a black cloth. For the collection, storage and visualization of f-NIRS data, the software application Oxysoft (Artinis Medical Systems, Zetten, The Netherlands) will be used. In addition to computing relative measurements of HBo2, HHb, and HbT using the modified Lambert Beer law, exportation of raw f-NIRS data will be performed and analyzed using a custom (MATLAB) script, used to inspect data for artifacts and processing.

Gait initiation (self-triggered) will be tested 3 times with each leg placed on a separate force platform according to the methodology employed in previous studies. Subsequently, the participants will walk barefoot along the 10 m laboratory walkway within ±5% of their preferred walking speed [24,25]. Trials will be repeated until at least 3 complete gait cycles are recorded with each foot making a clean contact on the force platforms located in the middle of the walkway.

### 2.9. Secondary Outcome Measures

Demographic and anthropometric measurements: age, sex, body mass index, marital status, educational level, profession, duration of the disease, medication and symptomatology. According to the WHO protocol (2016), body mass index (BMI) is classified as follows: thin (BMI < 18.5); eutrophic (BMI 18.5–24.9); overweight (BMI 25.0–29.9); pre-obesity and obesity (BMI > 30.0).

Unified Parkinson’s Disease Assessment Scale-III (UPDRS-III): The maximum value indicates greater involvement of the disease and the minimum normality. The UPDRS-III is a reliable (r-0.96) and valid scale.

The effectiveness of the synchronous learning format vs. the asynchronous remote video-based format will be based on the results of the aforementioned measurements and the response of patients who complete the protocol.

### 2.10. Statistical Analyses

The normal distribution of the dataset will be assessed using the Shapiro–Wilk test. Analyses of variance (ANOVAs) with repeated measures will be used to analyze kinematic and kinetic parameters and pre-frontal cortex hemodynamic responses during gait initiation and during the steady state between the two interventions and time points. Significant interactions and main effects will be further examined using Bonferroni post hoc analysis. The significance level will be set at *p* < 0.05.

## 3. Expected Results

The aim of the proposed study is to examine the acute effects of a MSK program on anxiety levels and select kinematic and kinetic parameters alongside DLPFC hemodynamics during gait initiation and steady-state walking in individuals with PD, performed in the following conditions: (a) synchronous learning format and (b) asynchronous remote video-based format. The program is customized for PD patients without deviating from the objectives of musicokinetic education. Primary outcomes will be the acute effects of a MSK protocol in people with PD on anxiety and locomotion. Secondary outcomes will be the effectiveness of asynchronous remote video-based format for those patients that are not able to attend classes on site. To our knowledge, this is the first time that MSK program will be introduced in PD as a form of exercise. Since motor dysfunctions are associated with anxiety [2], rhythmic-based interventions can potentially improve or even regulate it.

Anxiety may be a result of the neurochemical changes of the disease itself or a psychological reaction to the stress of the disease, and can be seen in various manifestations such as phobia, generalized anxiety disorder, or panic attacks [3,26]. The rationale behind the use of the purpose-designed MSK program for PD is its effectiveness in addressing the primordial motor expression of humans. Guidance by the rules of music, where periodicity is distinct, may eventually be beneficial in “calibrating” the brain, reducing motor difficulties, restoring movement harmony, and, thus, reducing anxiety. Moreover, we expect that the combined influence of improved rhythmicity, potentially improving the automatization of motor control, would be reflected in decreased DLPFC activation due to reduced executive load in the planned control of movements.

Given that recent research on rhythmic training has yielded beneficial results using methods such as rhythmic auditory stimulation (RAS) or dance, which includes rhythm as a key component, it is expected that the said MSK program will be an effective non-pharmacological intervention in dealing with PD-derived gait impairments. Finally, it is expected that both means, synchronous on site and tele format, will be equally effective in regulating the aforementioned biopsychological parameters as the content of the class and teaching styles will be the same. Research on the use of remote dance programs for people with PD has indicated acceptable feasibility and significant improvements in depression and quality of life, but at the same time has underlined the importance of live dance [27,28].

In general, patients with PD frequently experience disruptions in the 24 h daily profile of both behavioral and biological markers [29]. This lack of rhythmicity or disruption of inner-rhythm can negatively affect elements of health [30,31] and physical performance [32]. Therefore, it is rather prudent to establish non-pharmacological strategies able to reduce the global impact of the disease and improve the quality of life of both patients and the patient’s caregiver.

## Figures and Tables

**Table 1 medicina-59-01408-t001:** MSK for PD protocol.

Exercises	Description	Concept/Imagery	Goal
1. Warm up	Preparatory position: Sitting in a chair. Breathing with use of head and torso. Breathing with use of head, torso and arms. Repeat twice.	Instructions about how and when to inhale/exhale throughout movement.	To warm up To experience quarter notes
2. Warm up	Preparatory position: Sitting in a chair. Rising from the chair with use of head, torso and arms, travelling into high V and 5th position classical ballet. Slowly return to sitting position. Repeat 4 times. Finish upright position parallel legs, arms in 5th position. Repeat twice.	Clear instructions about how and when to inhale and exhale throughout movement	To warm up To experience quarter notes
3. Stretches	Preparatory position: Sitting in a chair. 8 slow side stretches with pauses. 16 quick side stretches with pauses Repeat the whole exercise.	Trying to reach something that is high	Flexibility and joint mobility. Half notes and quarters slow and quick tempo.
4. Stretches	Preparatory position: Sitting in a chair. 16 quick side stretches with pauses with a free pose at the end. Repeat the whole exercise.	Trying to reach something that is high.	Flexibility and joint mobility. Quarters—quick tempo.
5. Side bends	Preparatory position: Sitting in a chair. Arms high parallel position. 32 side bends with use of arms; left and right. Repeat the whole exercise.	“We are trees” Preference tree.	Flexibility, joint mobility, flow-smooth movement. Experiencing qualities of a waltz-3/4
6. Arm exercise	Preparatory position: Sitting in a chair. Arms near the torso bended, palms up—fist. Right arm—palm down figures open and to return to initial position Left arm—palm down figures open and return to initial position. Repeat 7 times. Repeat the whole exercise with palms up. Repeat the whole exercise.	Throwing colors; quick and sudden movements.	Power-arms Syncopation.
7. Arm exercise	Preparatory position: Sitting in a chair. Gathering all colors from the sky and take them near the torso, bend elbows and fist palms. 8 times, unfold and throw each arm in any direction. Repeat the whole exercise.	Fireworks, quick and sudden movement.	Experiencing Syncopation. Sharp movements.
8. Breathing exercise with sound	Preparatory position: Sitting in a chair Inhale from the nose arms going back, straight back pause Quick, sharp exhale from the mouth with wheezing, arms forward with curved back and syncopated movement pause. Repeat 3 times. Repeat whole exercise.	Breathing with soundwind whistling with breaks.	Experiencing whole notes and pauses via breathing.
9. Hand exercise	Preparatory position: Sitting in a chair Slightly bend elbows arms to the side circular movements of the wrists outward and inward.	Soft arms; like nobles of French royal court	Flexibility and dexterity—fine movements. Experiencing qualities of a waltz-¾.
10. Figure exercise	Preparatory position: Sitting in a chair. Figure trips. 4 times right hand and 4 times left hand. Repeat the whole exercise.	Soft and smooth movements; singing the counts	Dexterity. Experiencing qualities of a waltz-¾. counting and singing rhythm of the music.
11. Breathing exercise	Preparatory position: Sitting in a chair. Inhale with arms to the side slowly. Exhale quickly with wheezing and arms drop down on thighs curved back and pause. Repeat the whole exercise.	Slow and sustained movement vs. sharp and quick movement.	Experiencing whole notes. Slow sustained vs. sharp and quick movement.
12. Rhythm section	Claps, clips, taps and marches quarter notes and half notes. Repeat the whole exercise.	Understanding quarter and half notes.	Dexterity slight movement
13. Rhythm section	Preparatory position: Stand up legs parallel slightly open Walking steps through space any direction quarter notes and half notes quarter notes and half notes clapping on spot in any direction using personal space. Repeat the whole exercise.	Rhythm and movement using general and personal space for experiencing quarter and half notes.	Clapping quarters and half notes.
14. Center practice	Knee bends and stretches. Feet exercise, demi point and toes Rises with arms Claps Upward stretches Stretches Small and tall body Step touches with claps	Movement combinations based on previous exercises.	Flexibility Balance Coordination directions Use of personal and general space, experiencing half notes and quarter notes
15. Choreography	All taught movements combined in a small choreography. Repeat.	Dance steps	Application of rhythm and music into dance steps.
16. Cool down	Preparatory position sitting in a chair, arms on thighs. Head and arms exercises. Arm and back stretches, upward stretches breathing with arms.	Soft and flowing movements.	Experiencing qualities of adagio.

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
