# Peer review of "The Effects of a Single Session of a Rhythmic Movement Program on Selected Biopsychological Parameters in PD Patients: A Methodological Approach"

_medicina, 2023, doi:10.3390/medicina59081408_

Round 1
Reviewer 1 Report
The proposed study is interesting, however it presents limitations and aspects to correct:
The sample size is especially small, which conditions the possible results of the intervention.
The comparison is between two very similar interventions in synchronous or asynchronous modality, it would be convenient to include another group of control treatment with a conventional treatment, or justify the absence of the third group.
The study variables are new and it is a favorable point. However, the assessment of strength levels or an assessment of quality of life could be added.
The specific description of each exercise is well summarized in the table, but the time or repetitions of each step could be included. How was the intensity controlled?
The impact of the research is limited by not including the final results.
Author Response
Response to Reviewer 1 Comments
We would like to thank the reviewer for the comments. Please, find below our responses, highleted in red colour. Similarly, changes in the text can be found in red colour.
Please, note that the revised version of our manuscript contains also responses based on comments of all three reviewers.
Point 1: The sample size is especially small, which conditions the possible results of the intervention.
Response 1:
We would like to thank the reviewer for this comment. People with parkinson disease constitute a vulnerable group. Recruiting participants especially when the topic is rhythmic training can be challenging for several reasons. Indicatevely, one reason can be the accessiblity especially when patients experience mobility problems, or due to physical limitations or concerns about how they will be perceived by others.
Alhtough sample size seems small, it was estimated using the G power V 3.1.9.7 software (Heinrich-Heine-Universität, Düsseldorf, Germany). It has been found that the initial sample size required is 11. By considering a 10% dropout rate, 13 individuals with Parkinson’s disease will be recruited to voluntarily participate in this study. The above information is included in the manuscuscript in page 3, lines 123-129.
Point 2: The comparison is between two very similar interventions in synchronous or asynchronous modality, it would be convenient to include another group of control treatment with a conventional treatment, or justify the absence of the third group.
Response 2:
The suggested study design sounds interesting but at this stage our aim is not to compare musicokinetic program with something else but to examine its effectiveness on specific symtpomatology.
With regard to control group we have already explained above (response 1), the difficulty of recruiting patiens. This fact, led us not to include a control group but to adapt control situation using the same group of patients. In particular, all participants will undergo the intervention under the following conditions. The first time, participants will attend the intervention in the classroom, the second time, will attend the intervention remotely while, the thirt time participants will not attend any intervention. This will be the control situation. There will be a one-week period in between interventions where they will not engage in any activities. The said design does not require a third group, as we are examining three different conditions: in-class, (synchornous learning format), remotely (asynchronous remote-video) and no intervention.
However, for better clarity, we have added information regarding control situation in the methodology section in page 3, lines 139-141. “As control situation will be defined the data collection at the same time-points as above, but without any intervention following the 1-week washing out period”.
Point 3: The study variables are new and it is a favorable point. However, the assessment of strength levels or an assessment of quality of life could be added.
Response 3:
Thank you for your comment. Rhythmic or musicokinetic education is not a fitness platform [1]. Musicokinetic programmes focuse on the development of rhythmic awarness via movement. Synchronization and coordination as well as expression are important aspects of the said education.
We did not include any assessment of quality of life, as it is only a single session. Quality of life questionnaires include many questions that cannot be answered at this phase. For instance, there are questions about, sleeping, cutting their food, or carrying their groceries. Therefore, these questionnaires will not provide us with any meanful information regarding gait and anxiety levels. We hypothesise that changes will be minor and will be adequately detected through f-NIRS, motion analysis and the use of force platforms.
Point 4: The specific description of each exercise is well summarized in the table, but the time or repetitions of each step could be included. How was the intensity controlled?
Response 4:
As we already mentioned above, musicokinetic programmes are not fitness platforms. It is music and movement program. However, time is, indeed, included in the form of music terminology; for instance, whole notes, half notes quarter notes, etc. The dynamic (intensity) of each movement is associated with music, imagery and expression, which are included in the table (columns “concept/imagery” and “goals”).
Point 5: The impact of the research is limited by not including the final results.
Response 5:
We have no final results at the moment as the current work constitutes an experimental protocol, which will be conducted in the near future. According to MDPI intructions about protocols (https://www.mdpi.com/about/article_types) we have included “Expected Results”section instead.
References:
- Groves, W.C. Rhythmic Training and Its Relationship to the Synchronization of Motor-Rhythmic Responses. Journal of Research in Music Education 1969, 17, 408-415, doi:10.2307/3344169.
Reviewer 2 Report
This cross-over randomized study was aimed to investigate the effects of musicokinetic therapy in patients with parkinson. General, this study had not been carried out carefully, there are several problems.
Major issues
I did not see the result, discussion and conclusion parts of the manuscript so ı did not complete my review. Please add this missed parts.
Minor Issues
Abstract
Please delete the novelty statement. In general, the abstract need rewrite by the authors base on the rules of the journal and findings.
Please use proper keywords from MESH, and don't use the abbreviation.
Introduction
The author need to mention the validity of f-NIRS in neurodegenerative diseases.
Participants
Please add fullwritten of MDS, and use the abbreviations with full written when they used first time.
Material and Method
Please add a flow chart and report the excluded patients, we just know 13 patients attended the study.
The authors can give the table 1 as a appendix.
The quality of English language can improve.
Author Response
Response to Reviewer 2 Comments
We would like to thank the reviewer for the comments. Please, find below our responses, highleted in red colour. Similarly, changes in the text can be found in red colour.
Please, note that the revised version of our manuscript contains also responses based on comments of all three reviewers.
Major Issues
Point 1: I did not see the result, discussion and conclusion parts of the manuscript so ı did not complete my review. Please add this missed parts.
Response 1:
With all due respect, we would like to explain that this article is not a study but a protocol which, has not been implemented yet. Therefore, sections such as results, discussion and conclusion cannot be encompassed. According to MDPI guilines for protocols (https://www.mdpi.com/about/article_types) we should include: Abstract, Keywords, Introduction, Experimental Design, Materials and Equipment, Detailed Procedure, and Expected Results. All the aforementioned parts have been incorporated in the current work.
Minor issues
Point 2: Abstract. Please, delete the novelty statement. In general, the abstract need rewrite by the authors base on the rules of the journal and findings.
Response 2:
Novelty statement has been deleted from the text. Please, see page 1, lines 31-32. The text now reads as follows: “The combined implementation of f-NIRS in parallel with 3D gait analysis, has yet to be evaluated in Parkinsonian patients following a MSK intervention”.
Since the current work is a protocol article (please comment above), no other changes have been introduced to the rest of the abstract.
Point 3: Introduction. The author needs to mention the validity of f-NIRS in neurodegenerative diseases.
Response 3:
We would like to thank the reviewer for this comment. The validity of f-NIRS have been added in the introduction, page 3, lines 96-102. The text now reads as follows: “fNIRS is extensively employed in chronic neurological disorders associated with aging, enabling direct measurements of the absolute concentrations of oxygenated and deoxygenated hemoglobin in the brain [16]. In relation to Parkinson's disease, fNIRS investigates the activity of the prefrontal cortex during tasks such as gait, postural stability, and dual-task performance thereby determining motor vs. non-motor effects and executive dysfunction due to the disease’s pathology [16].”
Point 4: Participants. Please, add fullwritten of MDS, and use the abbreviations with full written when they used first time.
Please, use proper keywords from MESH, and don't use the abbreviation.
Response 4:
We would like to thank the reviewer for bringing this to our attention. Full words of MDS can be found in the text in page 3, lines 110-111.
“…to Movement Disorders Society (MDS) clinical diagnostic..”
Also proper keywords from MESH have been added in the text. Please, see page 1, line 37.
“dorsolateral prefrontal cortex”.
Point 5: Material and Method. Please, add a flow chart and report the excluded patients, we just know 13 patients attended the study.
Response 5:
This is a very good point however, recruitment has yet to start and therefore it is not possible to include a flow chart.
Point 6: The authors can give the table 1 as a appendix.
Response 6:
As explained above, this work is a protocol. According to MDPI guilines for protocols (https://www.mdpi.com/about/article_types) we should include: Abstract, Keywords, Introduction, Experimental Design, Materials and Equipment, Detailed Procedure, and Expected Results. There is no mention of appendix. Besides, as the proposed research will be conducted in the near future, we believe that it is crucial not to include the content of the protocol as an appendix.
Reviewer 3 Report
Dear Author
The purpose of this valuable article is to examine the acute effects of a specially designed MSK program on anxiety levels, frontal cortical hemodynamic responses during gait initiation and steady-state walking, in patients with Parkinson's disease.
On the other hand, the description of risk management regarding the participants is unmeasured, which is an important item when teaching at a distance, since PD is also at high risk for falls. Therefore, it should be mentioned regarding risk management during the intervention in the method.
Author Response
Response to Reviewer 3 Comments
We would like to thank the reviewer for the comments. Please, find below our responses, highleted in red colour. Similarly, changes in the text can be found in red colour.
Please, note that the revised version of our manuscript contains also responses based on comments of all three reviewers.
Point 1: The purpose of this valuable article is to examine the acute effects of a specially designed MSK program on anxiety levels, frontal cortical hemodynamic responses during gait initiation and steady-state walking, in patients with Parkinson's disease.
On the other hand, the description of risk management regarding the participants is unmeasured, which is an important item when teaching at a distance, since PD is also at high risk for falls. Therefore, it should be mentioned regarding risk management during the intervention in the method.
Response 1:
We would like to thank the reviewer for bringing this important aspect to our attention. Risk management strategy has been added in the methodology section in page 4, lines 159-162. The relevant text now reads as follows “Risk management strategy includes a qualified carer who will be present during both modalities synchronous and asynchronous, full explanation of the experimental procedures, controlled laboratory temperatures at about 22o Celsius, as well as the continuous presence of at least two members of the research team.”
Round 2
Reviewer 2 Report
Thank you for your corrections, I missed your study type at the first of my review. I read your revised text on the light of this perspecitve again and your answers are quite well informative for my comments.
My suggestion to authors before the publication, they can write parkinson disease instead of "PD" in the title.